# RNA chaperones buffer deleterious mutations in *E. coli*

**Marina Rudan[1], Dominique Schneider[2,3], Tobias Warnecke[4]\*[†], Anita Krisko[1]\*[†]**

[1]Mediterranean Institute for Life Sciences, Split, Croatia; [2]Laboratoire Adaptation et Pathogénie des Microorganismes, Université Grenoble Alpes, Grenoble, France; [3]Centre National de la Recherche Scientifique, Grenoble, France; [4]Molecular Systems Group, MRC Clinical Sciences Centre, Imperial College, London, United Kingdom

**Abstract** Both proteins and RNAs can misfold into non-functional conformations. Protein chaperones promote native folding of nascent polypeptides and refolding of misfolded species, thereby buffering mutations that compromise protein structure and function. Here, we show that RNA chaperones can also act as mutation buffers that enhance organismal fitness. Using competition assays, we demonstrate that overexpression of select RNA chaperones, including three DEAD box RNA helicases (DBRHs) (CsdA, SrmB, RhlB) and the cold shock protein CspA, improves fitness of two independently evolved *Escherichia coli* mutator strains that have accumulated deleterious mutations during short- and long-term laboratory evolution. We identify strain-specific mutations that are deleterious and subject to buffering when introduced individually into the ancestral genotype. For DBRHs, we show that buffering requires helicase activity, implicating RNA structural remodelling in the buffering process. Our results suggest that RNA chaperones might play a fundamental role in RNA evolution and evolvability.

**\*For correspondence:** tobias. warnecke@csc.mrc.ac.uk (TW); anita.krisko@medils.hr (AK)

[†]These authors contributed equally to this work

**Competing interests:** The authors declare that no competing interests exist.

**Reviewing editor**: Naama Barkai, Weizmann Institute of Science, Israel

## Introduction

Protein chaperones can buffer the effects of mutations that affect protein stability and/or folding, as evidenced by the release of cryptic genetic variation upon inhibition of Hsp90 (*Rutherford and Lindquist, 1998*; *Queitsch et al., 2002*; *Rohner et al., 2013*), increased enzyme evolvability in *Escherichia coli* strains overexpressing GroEL (*Tokuriki and Tawfik, 2009*), accelerated rates of evolution in habitual chaperone clients (*Bogumil and Dagan, 2010*; *Warnecke and Hurst, 2010*; *Williams and Fares, 2010*), lower mutational penetrance in *Caenorhabditis elegans* larvae with higher Hsp90 titres during embryonic development (*Burga et al., 2011*), and increased fitness of *E. coli* mutator strains following GroEL overexpression (*Fares et al., 2002*). Whether RNA chaperones play a similarly pervasive role in buffering mutations that affect RNA structure or folding, however, has not been addressed empirically.

RNA misfolding is common (*Herschlag, 1995*) and frequently produces long-lived alternate structures (*Downs and Cech, 1996*) that require the assistance of RNA-binding proteins for timely resolution. Like their protein-folding counterparts, RNA chaperones can promote orderly structural transitions towards and subsequently stabilize the native fold or—as exemplified by classic work on the *Neurospora crassa* CYT-19 protein (*Mohr et al., 2002*; *Bhaskaran and Russell, 2007*)—facilitate the refolding of misfolded species (*Russell, 2008*). Of particular interest in this regard are DEAD box RNA helicases (DBRHs), which can alleviate folding errors by unwinding short RNA helices, thus enabling renewed exploration of the folding landscape (*Pan and Russell, 2010*). By the same token, DBRH activity may also counteract mutations that precipitate a deleterious increase in RNA stability. Many DBRHs, including RhlB (a component of the *E. coli* degradosome [*Py et al., 1996*]) and the eukaryotic translation initiation factor eIF4A are involved in the folding and unfolding of diverse RNAs and might therefore act as broad-spectrum mutation buffers.

**eLife digest** Stretches of DNA known as genes contain the instructions to make the proteins and RNA molecules that are essential for life. The DNA sequence of the gene is first copied to make a strand of RNA, which may subsequently be 'translated' to make a protein. To carry out their tasks, proteins and many RNA molecules must fold into specific three-dimensional structures. Since folding can easily get derailed, proteins known as chaperones assist with this process.

Mutations sometimes occur in the DNA that reduce the ability of the proteins or RNA molecules to fold correctly. Previously, scientists had found that some chaperones help incorrectly folded proteins adopt 'normal' shapes and thus mask the harmful effects of mutations. However, it was not known whether the chaperones that fold RNA similarly suppress harmful mutations.

To address this question Rudan et al. studied the effects of several RNA chaperones in *Escherichia coli* bacteria that had been grown in the laboratory as part of long-term evolution experiments. During this time, they had accumulated mutations that reduced the fitness of later generations in comparison with their ancestors. Rudan et al. then found that increasing the levels of certain RNA chaperones—in particular, a group called DEAD box RNA helicases—in the evolved bacteria improved their fitness. This strongly suggests that RNA chaperones, like protein chaperones, can suppress harmful mutations. Compromised versions of the same RNA chaperones, which were unable to dismantle folded RNA structures, did not show any improvements in fitness, demonstrating that the capacity to unfold and refold RNA is critical.

Rudan et al. suggest that different types of chaperones are likely to alleviate RNA mutations using different mechanisms. A future challenge will therefore be to work out how these mechanisms work together to mask different mutations and allow them to persist through evolution, their harmful effects rendered invisible to the forces of natural selection.

## Results

### Overexpression of DBRHs enhances fitness of a low-fitness strain

We tested whether DBRHs buffer deleterious mutations in vivo using a fitness rescue paradigm (*Fares et al., 2002*). Briefly, an *E. coli* strain propagated under conditions of weak selection is expected to accumulate deleterious mutations and experience a concomitant decline in fitness compared with its ancestor. Similarly, strains adapting to a novel environment will accumulate not only beneficial but also deleterious mutations, which may hitchhike along with beneficial alleles and can be exposed as deleterious in the old environment (*Fares et al., 2002*). In both scenarios, buffering can be inferred if (over)expression of a candidate chaperone leads to a greater fitness gain in the low-fitness evolved strain compared with its ancestor. We therefore performed pairwise competition experiments (*Lenski, 1991*) between the *E. coli* REL606 strain, which is the ancestor of the long-term evolution experiment (LTEE), and two evolved *mutS* mutator strains that were sampled from a lineage after ~20,000 (20k) and ~40,000 (40k) generations of adaptation to a minimal glucose-limited medium (*Sniegowski et al., 1997*) (*Figure 1A*). Performing 24-hr competitions in the alternative LB medium, we observed reduced fitness relative to the REL606 ancestor for the 40k but not for the 20k genotype (*Figure 1B*). Reduced fitness in the 40k strain cannot be proximately attributed to the *mutS* mutation, which underlies the mutator phenotype; this mutation arose ~3000 generations into the LTEE (*Sniegowski et al., 1997*) and is already present in the 20k strain. Thus, the mutation(s) responsible for reduced fitness emerged later.

Having established the presence, in the 40k strain, of deleterious mutations potentially amenable to buffering, we introduced plasmids carrying a specific *E. coli* DBRH gene (either *csdA*, *rhlB*, or *srmB*) into each of the three genetic backgrounds (ancestor, 20k, and 40k). We then competed each transformed strain against a strain of the same genotype but bearing an empty control plasmid. Whereas overexpression in the ancestral and 20k backgrounds had limited effects on competitive fitness, overexpression of either DBRH enhanced fitness of the mutationally compromised 40k genotype (*Figure 1C*). For each DBRH tested, fitness gains were abolished when we introduced mutations that rendered the respective helicase domain catalytically inactive (*Figure 1C*), suggesting that helicase and therefore RNA remodelling activities are essential for buffering.

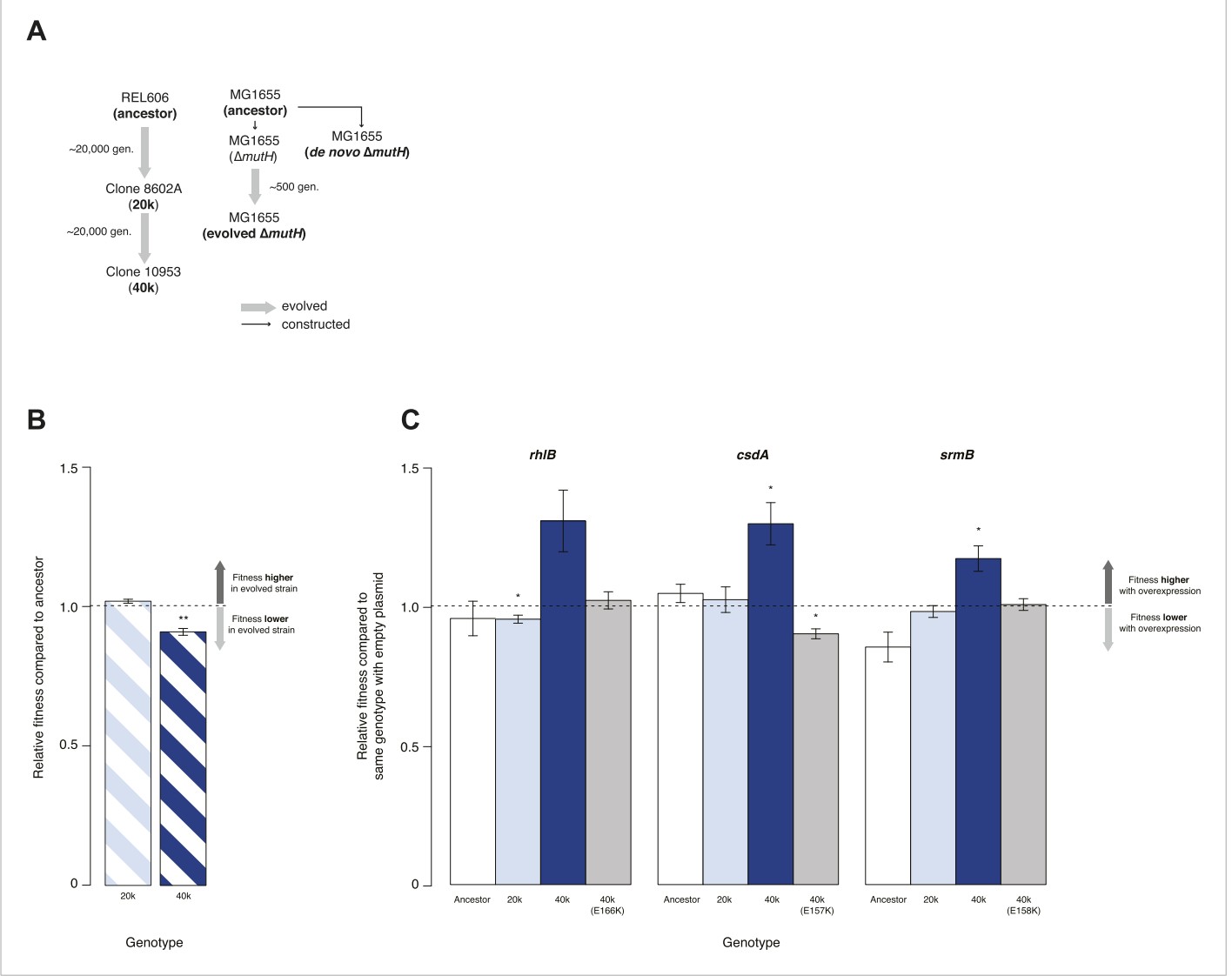

**Figure 1**. Relative fitness of *Escherichia coli* REL606-derived strains. (**A**) Relationships between strains used in different competition assays. Short names of competed strains are given in bold; gen.: generations. (**B**) Relative fitness of the 20k and 40k genotypes, each competed against their REL606 ancestor. (**C**) Relative fitness of ancestral and evolved genotypes overexpressing one of three DEAD box RNA helicases (DBRHs) compared with identical strains carrying the empty control plasmid. E166K, E157K, and E158K: competitions in the 40k background where plasmids carried mutated versions of the respective DBRH. In each case, the central glutamic acid residue of the DEAD motif has been recoded to lysine, compromising the helicase activity. Bar heights indicate mean relative fitness across four biological replicates, with each mean derived by averaging over four technical replicates. Error bars represent standard errors of the mean. **p < 0.01, *p < 0.05 (one-sample *t*-test). Additional results for competitions terminated in mid-exponential phase (after 2 hr) are shown in *Figure 1—figure supplement 1*.

The following figure supplement is available for figure 1:

**Figure supplement 1**. Relative fitness in competition experiments terminated in mid-exponential phase (REL606).

## RNA chaperones buffer distinct mutations in a second low-fitness strain

Protein chaperones like GroEL and Hsp90 target misfolded substrates by recognizing exposed hydrophobic patches that are buried in the native state (*Hartl et al., 2011*). This generic mechanism allows buffering to occur across a broad range of substrates, differentiating these chaperones from gene- or pathway-specific suppressors. To determine whether DBRH-mediated buffering encompasses diverse target substrates, we performed the same suite of fitness rescue experiments in

a second, independently evolved strain. This MG1655-derived *mutH* deletion (Δ*mutH*) strain was sampled after a shorter period of laboratory evolution (∼500 generations) but also exhibited reduced fitness compared with its ancestor (*Figure 2A*) and experienced fitness gains upon DBRH overexpression (*Figure 2B*). We confirmed that fitness effects were not directly related to the *mutH* deletion by deleting *mutH* in the ancestral MG1655 background de novo. Fitness of the de novo Δ*mutH* strain was not reduced compared with the deletion-free ancestor (*Figure 2A*).

To rule out buffering of identical mutations across strains, we sequenced the genomes of the evolved Δ*mutH* strain and its laboratory ancestor. As expected, we found significantly fewer mutations in the evolved Δ*mutH* compared with the 40k strain (*Table 1*, *Supplementary files 1–3*). More importantly, there were no identical point mutations or indels in the two evolved strains, implying buffering of independent mutations. This might be indicative of a general rather than gene- or pathway-specific buffering mechanism and is consistent with DBRHs being broad-spectrum catalysts of RNA remodelling that recognize and target misfolded substrates through a non-specific mechanism of action (*Jarmoskaite et al., 2014*).

## Pinpointing individual deleterious mutations buffered by RNA chaperones

To lay the groundwork for unravelling the molecular basis of buffering, we sought to identify mutations that are individually deleterious and whose effects on fitness are ameliorated by chaperone overexpression. In the absence of strong a priori candidates for DBRH-mediated buffering—neither strain harboured mutations in known structural RNAs (*Supplementary files 2, 3*)—the evolved Δ*mutH* strain, carrying only 12 point mutations (*Table 1*), affords us the rare opportunity to pinpoint such mutations systematically.

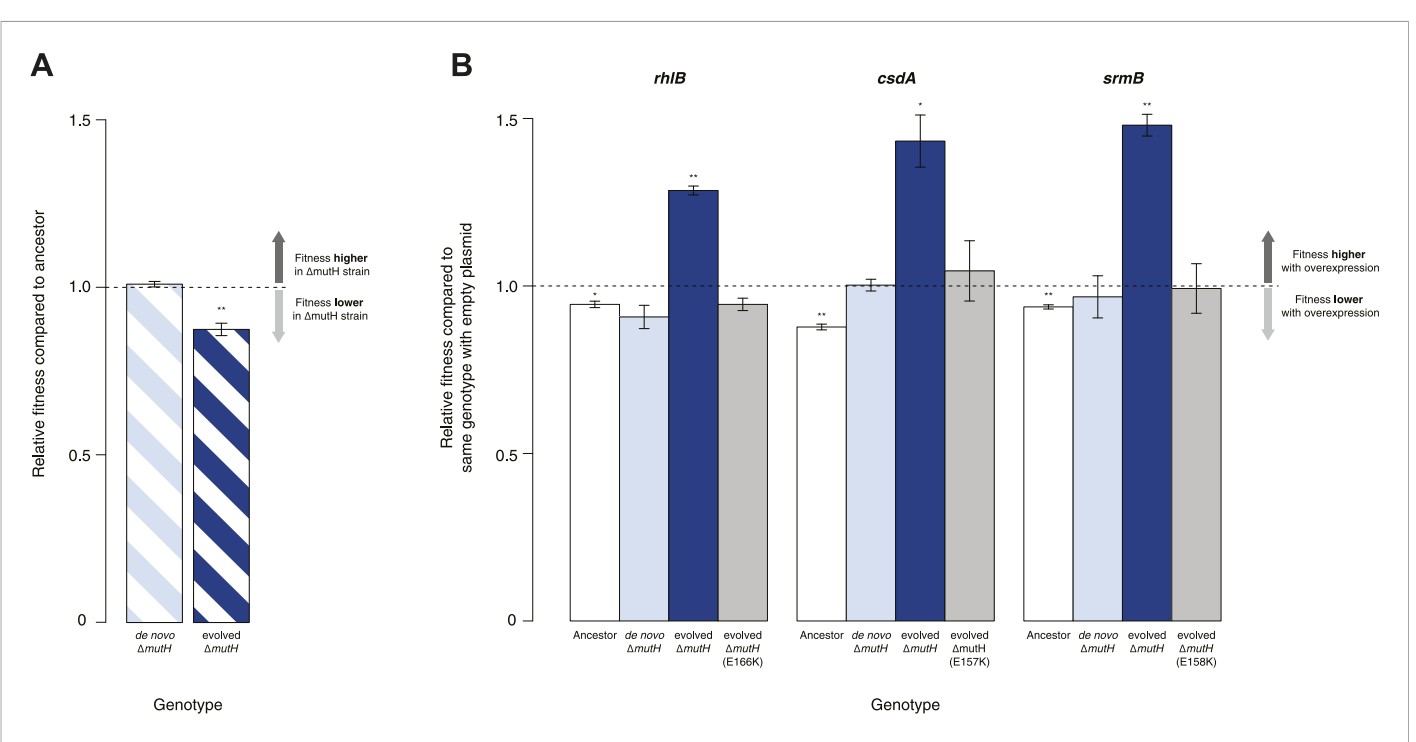

**Figure 2**. Relative fitness of *Escherichia coli* MG1655-derived strains. (**A**) Relative fitness of the evolved and de novo-constructed Δ*mutH* strains, each competed against their MG1655 ancestor. (**B**) Relative fitness of ancestral, evolved, and de novo Δ*mutH* genotypes overexpressing one of three DEAD box RNA helicases compared with identical strains carrying the empty control plasmid. E166K, E157K, and E158K, bar heights and error bars are as described in *Figure 1*. **$p < 0.01$, *$p < 0.05$ (one-sample *t*-test). Additional results for competitions terminated in the mid-exponential phase (after 2 hr) are shown in *Figure 2—figure supplement 1*.

The following figure supplement is available for figure 2:

**Figure supplement 1**. Relative fitness in competition experiments terminated in mid-exponential phase (MG1655).

**Table 1.** Number of mutations in evolved mutator strains compared with their respective ancestors

| Strain | SNPs (CDS/total)* | Small indels† (CDS/total) | Large deletions |
|---|---|---|---|
| 20k | 667/755 | 86/129 | 1 |
| 40k | 1163/1291 | 128/183 | 4 |
| Evolved Δ*mutH* | 12/12 | 0/2 | 0‡ |

*CDS, coding sequence; SNP, single-nucleotide polymorphism.
†≤4 bp.
‡Excluding the *mutH* deletion itself.

Using a recombineering approach (see 'Materials and methods'), we individually introduced each mutation located in a gene of known function (N = 7, five mutations are located in y-genes) into the MG1655 genome. We then competed each of these strains, isogenic except for a single mutation, against MG1655. Only one strain, carrying a mutation in the *lamB* gene, exhibited both reduced fitness and evidence for buffering (*Figure 3A*). Although primarily known for its role in maltose uptake, *lamB* is a more general glycoporin that becomes derepressed under glucose-limiting conditions to maximize sugar uptake (*Death et al., 1993*). *LamB* deletion mutants are outcompeted by reference strains when grown on glucose (*Death et al., 1993*), suggesting a possible cause of fitness loss in our strain. Growth phase-related fitness patterns were mirrored by the evolved Δ*mutH* strain (*Figure 2—figure supplement 1*) consistent with *lamB* being a dominant driver of fitness loss in this strain.

For the 40k strain, where the large number of mutations precludes comprehensive analysis, we focused on two mutations in the essential ribosomal protein gene *rplS*—one synonymous (*rplS^syn^*) and one non-synonymous (*rplS^nonsyn^*, *Supplementary file 2*). Both mutations are present in the 40k but not in the 20k strain. As demonstrated for *rplA* and *rpsT* in *Salmonella typhimurium* (*Lind et al., 2010*), synonymous mutations in ribosomal protein genes can strongly compromise fitness, indicating the presence of selective constraints unrelated to amino acid composition. Moreover, fitness costs of mutations in *rplA/rpsT* correlated with changes in predicted mRNA free energy, albeit weakly (*Lind et al., 2010*). We therefore performed competitions between the REL606 ancestor and strains carrying either the *rplS^syn^* or *rplS^nonsyn^* mutation. These competitions revealed a deleterious effect of *rplS^syn^*, buffered by DBRH overexpression (*Figure 3A*). In contrast to *lamB*, growth phase-specific buffering patterns in *rplS^syn^* did not echo observations in the 40k strain (*Figure 3—figure supplement 2*; *Figure 1—figure supplement 1*), consistent with the presence of multiple fitness-relevant mutations in the latter.

These results establish that RNA chaperones buffer individual deleterious mutations although the mechanisms of buffering remain, at this point, unresolved. We consider likely mechanisms of buffering and appropriate experimental follow-ups in the 'Discussion'.

## Fitness gains upon overexpression of cspA suggest diverse mechanisms of buffering

Buffering of mutations that affect RNA structure and folding may be mechanistically diverse, rather than limited to a DBRH model where helicase activity catalyses the local rupture of helices and enables structural remodelling. To explore mechanistic diversity in buffering, we considered the cold shock protein CspA. By binding with low specificity to single-stranded RNA, CspA can prevent the formation of unwanted secondary structure and thereby narrow the RNA folding landscape (*Jiang and Hou, 1997*)—a mechanism of action reminiscent of the ubiquitous protein chaperone DnaK (Hsp70), which cycles on and off nascent polypeptide chains to allow ordered stepwise folding (*Hartl et al., 2011*). We found that overexpression of CspA is associated with fitness gains in both low-fitness genotypes but not in the corresponding ancestral strains (*Figure 4*). By contrast, overexpression of a mutant version of CspA with severely reduced nucleic acid-binding activity (*Hilier et al., 1998*) did not confer fitness benefits upon overexpression (*Figure 4*). Buffering occurs although CspA levels are relatively low compared with overexpressed DBRHs (~fourfold and ~twofold reduced relative abundance compared with CsdA and RhlB/SrmB, respectively, *Figure 5*), likely because CspA is subject to negative autoregulation (*Bae et al., 1997*).

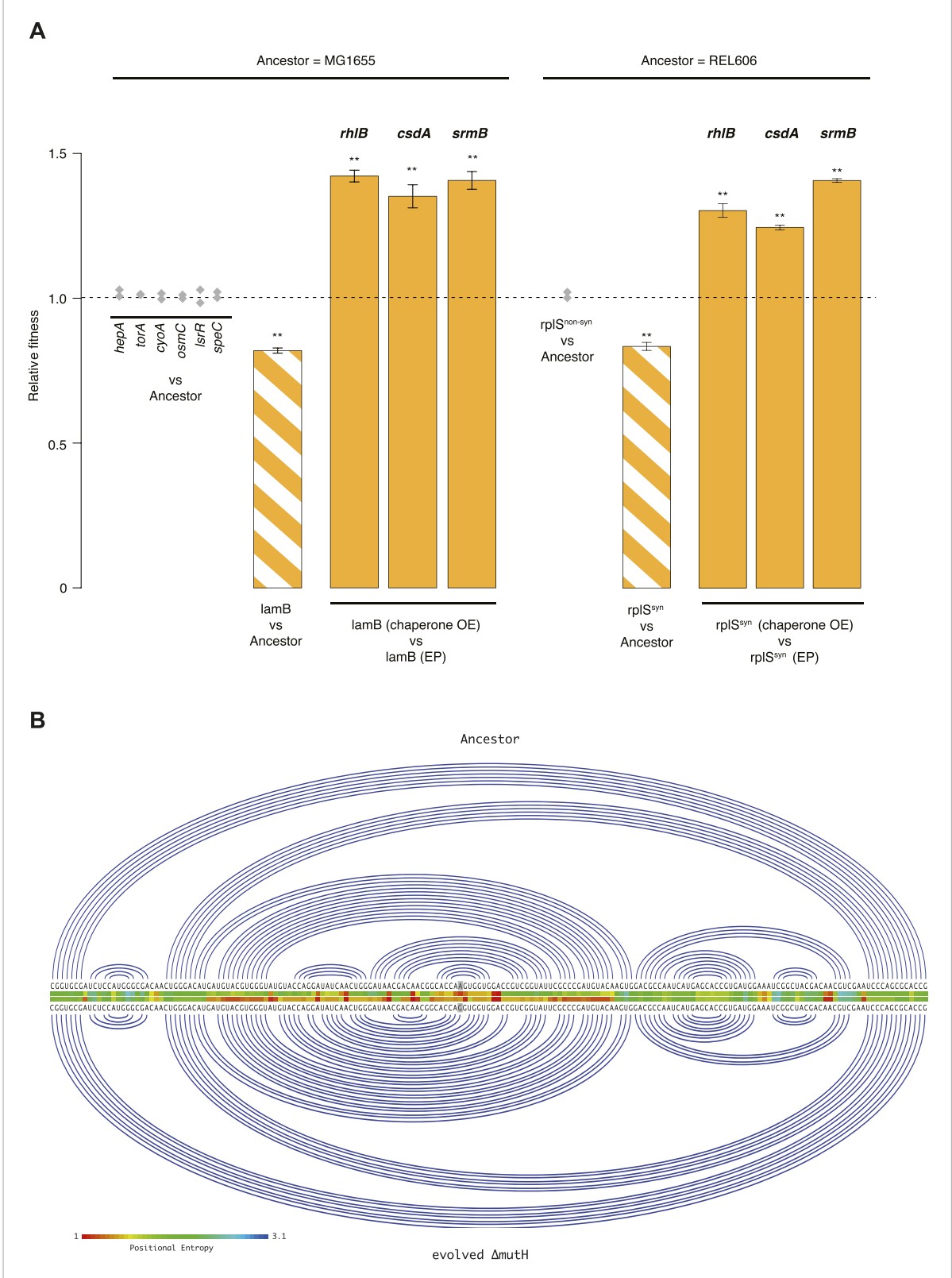

**Figure 3**. Fitness effects and buffering of individual mutations. (**A**) Relative fitness of strains carrying single point mutations introduced into the relevant ancestral background competed against the respective ancestor. The mutations correspond to those listed in **Supplementary files 2, 3** for the respective genes. Initial screening for fitness defects involved two biological replicates (grey diamonds). For the two mutations where the initial screen suggested

*Figure 3. continued on next page*

*Figure 3. Continued*

a measurable fitness deficit, *lamB* and *rplS*$^{syn}$, all competitions were carried out in quadruplicate. Bar heights and error bars are as described in *Figure 1*. **p < 0.01, *p < 0.05 (one-sample *t*-test). OE: overexpression; EP: empty plasmid. Additional results for competitions terminated in mid-exponential phase (after 2 hr) are shown in *Figure 3—figure supplements 1, 2*. (**B**) Linear Feynman graph of the *lamB* region that harbours the mutation in the evolved Δ*mutH* strain (highlighted in grey). We predicted RNA secondary structure for the entire *malK-lamB-malM* transcription unit (RegulonDB identifier: ECK120009315) and its mutated counterpart using RNAfold (*Lorenz et al., 2011*). The *malK-lamB-malM* operon contains a repetitive extragenic palindromic (REP) element downstream of *lamB*, which prevents premature degradation of the *lamB* cistron following cleavage from *malM*. Resolution of this REP element as part of regulated degradation was previously shown to require RhlB (*Khemici and Carpousis, 2004*). However, comparison of predicted minimum free energy structures between wild type and mutant *malK-lamB-malM* transcripts suggested structural changes that do not interfere with REP element formation but rather lead to decreased positional entropy at the local level, as highlighted here.

The following figure supplements are available for figure 3:

**Figure supplement 1**. Relative fitness in competition experiments terminated in mid-exponential phase (*lamB*).

**Figure supplement 2**. Relative fitness in competition experiments terminated in mid-exponential phase (*rplS*$^{syn}$).

These results show that buffering can be mediated by RNA chaperones that interact with their substrates in mechanistically distinct ways. Interestingly, cold shock proteins and DBRHs have been suggested to work together, with helicases opening double-stranded structures and cold shock proteins binding to the single-stranded end products, their combined activities preventing the formation of unfavourable structures (*Hunger et al., 2006*). Overexpression of different RNA chaperones, including the proteins considered here, might therefore benefit identical RNA target species.

## Discussion

Elucidating in greater detail how different chaperones interact with wild-type and mutant RNAs will be critical to elucidate the molecular basis of buffering and how it relates to altered RNA secondary and tertiary structure. Here, we demonstrate that buffering by RNA chaperones occurs at the organismal level, establish that helicase and nucleic acid-binding activity are required for buffering by DBRHs and CspA, respectively, and identify individual mutations that are amenable to buffering. These mutations, located in *lamB* and *rplS*, constitute valuable assets to establish the precise molecular mechanism(s) of buffering in the future.

Although this study was not designed to resolve molecular mechanism, it is nonetheless useful, if primarily to guide future work, to contemplate potential causes of fitness loss, and how buffering might occur. In particular, we wanted to know whether mutations in *lamB* and *rplS* stand out amongst other mutations in their predicted effect on RNA structure. Several metrics that quantify mutational impact on RNA secondary structure do indeed suggest comparatively severe effects for *rplS*$^{syn}$ (e.g., the correlation of base pairing probabilities, *Supplementary file 2*). Similarly, of all the mutations in the evolved Δ*mutH* strain, the *lamB* mutation is predicted to have the most severe repercussions for local RNA structure as measured by maximum local base pair distance ($d_{max}$, *Supplementary file 3*, *Figure 3B*). However, these correlates should be interpreted with caution. Existing measures of RNA structural change, however accurate, will only indicate how disruptive a given mutation is to the RNA structure but not whether the resulting defect can be rescued by chaperone activity, or whether it matters at the organismal level. Consequently, we would not necessarily anticipate a robust relationship between fitness loss, buffering, and indicators of RNA structural change. More mundanely, we cannot quantify the reliability of any (structural) predictor without a larger set of experimentally characterized mutations that are both deleterious and amenable to buffering. In short, while suggestive, these findings do not conclusively implicate RNA structure as the vehicle for fitness loss.

Provided deleterious effects arise at the level of RNA structure, RNA chaperone activity might be beneficial through stabilizing (CspA), destabilizing, or remodelling (DBRHs) affected structures in the focal transcript or through changing how these transcripts interact with other RNAs or RNA-binding proteins (*Pan and Russell, 2010*). Considering *lamB*, one possibility is that an increase in local stability

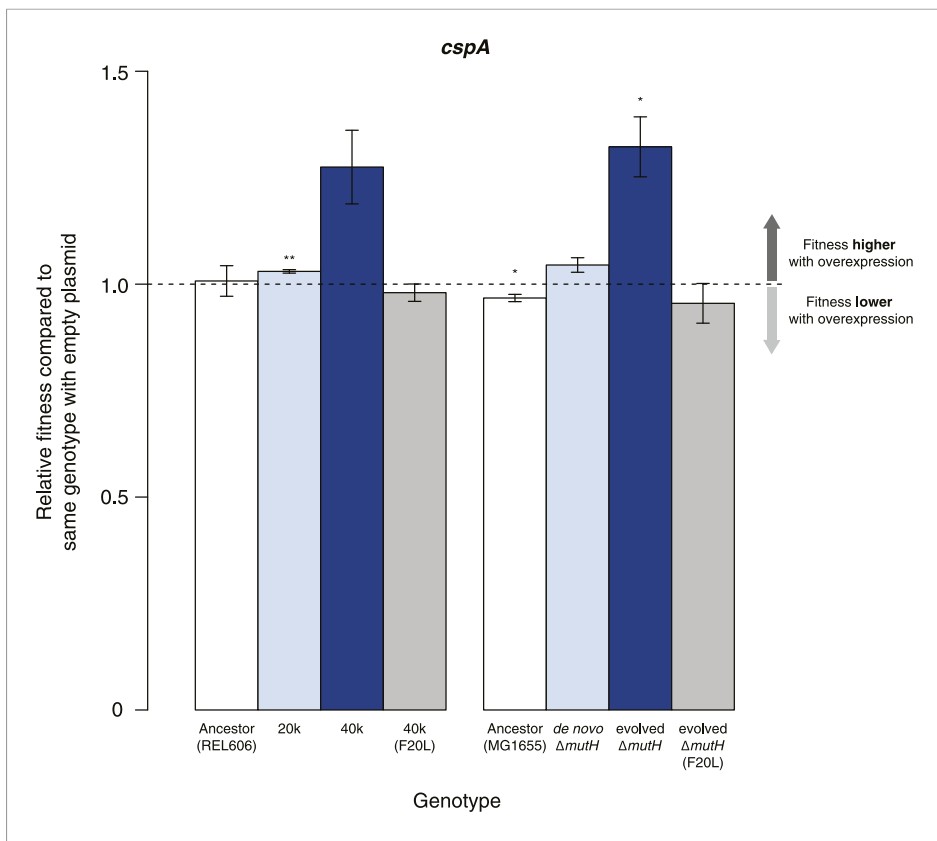

**Figure 4**. Effects of CspA overexpression on relative fitness. Relative fitness of REL606- and MG1655-derived strains overexpressing CspA compared with strains of the same genotype carrying the empty control plasmid. F20L: competitions in the 40k and evolved Δ*mutH* backgrounds, respectively, where plasmids carried a mutated version of the *cspA* gene yielding a protein with compromised nucleic acid binding ability (*Hilier et al., 1998*). Bar heights and error bars are as described in *Figure 1*. **p < 0.01, *p < 0.05 (one-sample *t*-test). Additional results for competitions terminated in mid-exponential phase (after 2 hr) and competitions involving the *lamB* mutant in the evolved Δ*mutH* strain and the *rplS*^syn mutant in the 40k strain are shown in *Figure 4—figure supplement 1*.

The following figure supplement is available for figure 4:

**Figure supplement 1**. Relative fitness in competition experiments terminated in mid-exponential phase (*cspA*).

caused by the mutation (*Figure 3B*) introduces a translational roadblock that is resolved by RNA chaperone activity. Alternatively, one might envisage a more complex scenario, where the non-synonymous *lamB* mutation gives rise to a dominant negative protein product and RNA chaperone overexpression ameliorates fitness defects by facilitating degradation of the mRNA, thus reducing levels of mutant LamB protein. Both of these mechanisms are speculative, and endorsing either one (or a third or fourth option) would be premature. Rather, competing mechanistic hypotheses will have to be confirmed or debunked through targeted follow-up experiments. For example, a logical first step to eliminate one of the hypotheses above would be to measure LamB levels (predicted to increase and decrease upon chaperone overexpression, respectively). In designing insightful follow-up studies, a few issues deserve wider consideration.

First, buffering might be direct (involving interactions between the RNA chaperones and the mutant RNA) or indirect (involving interactions between the RNA chaperone and other components of the cell, which in turn lead to buffering). In addition to dissecting specific chaperone–RNA interactions, it will therefore be important to establish system-level effects of RNA chaperone overexpression.

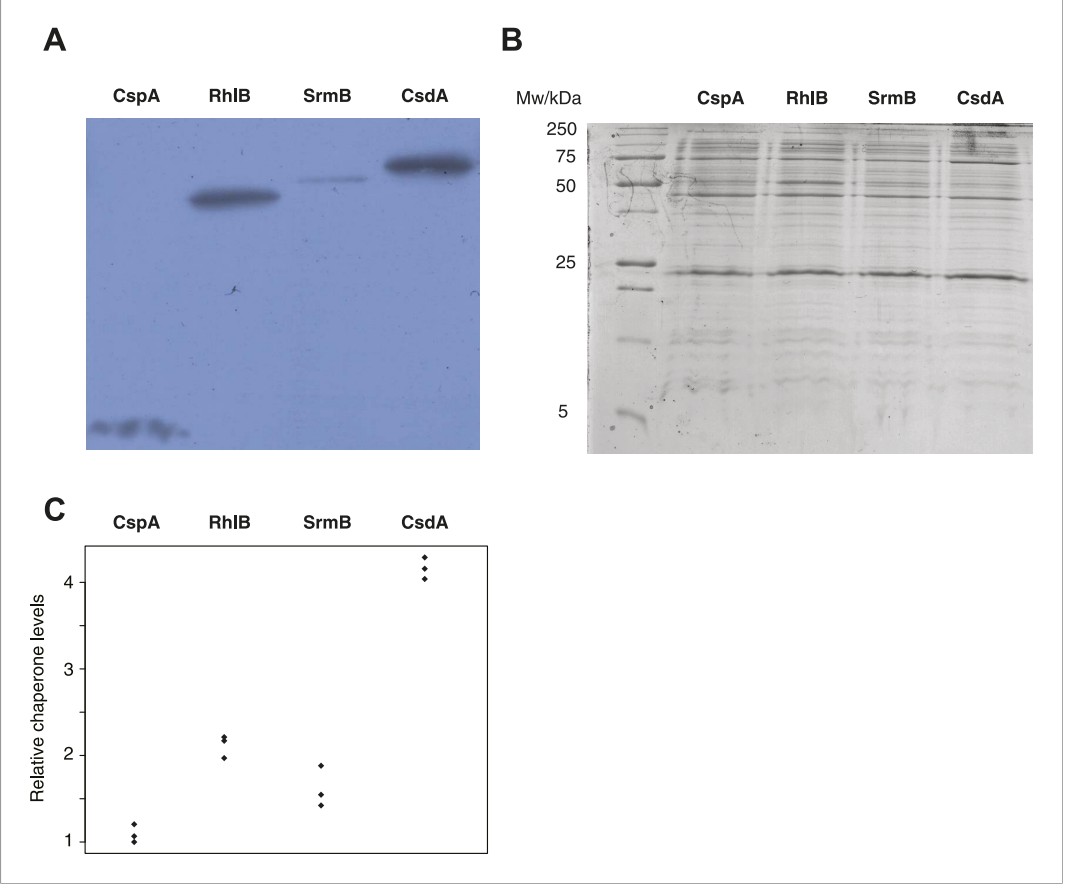

**Figure 5**. Relative chaperone abundances. (**A**) Representative Western blot for evolved Δ*mutH* strains overexpressing one of the focal RNA chaperones. Molecular weights (from nucleotide sequence): CspA, 7.403 kD; RhlB, 47.126 kD; SrmB, 49.914 kD; CsdA, 70.546 kD. (**B**) Representative Coomassie-stained SDS-PAGE gel. (**C**) Relative chaperone levels are defined as the ratio of Western blot intensity to Coomassie intensity (see 'Materials and methods'). The lowest ratio detected across triplicate experiments in all strains was set to one. Comparing these ratios between strains overexpressing different RNA chaperones gives a semi-quantitative indication of relative chaperone abundances. For example, CsdA levels in CsdA-overexpressing cells are ~fourfold higher than CspA levels in CspA-overexpressing cells. Note that this metric does not allow conclusions about the absolute fraction of total protein that is occupied by each chaperone in the different strains.

Second, both synonymous and non-synonymous mutations can affect RNA structure so that non-synonymous mutations (like the one found in *lamB*) should not be discarded a priori as unlikely candidates for buffering by RNA chaperones; deleterious consequences might arise at the RNA level even though the amino acid change is selectively neutral. Conversely, synonymous mutations can affect translation kinetics and protein folding (*Plotkin and Kudla, 2011*) and therefore have fitness repercussions at the protein level. As a corollary, some mutations might be amenable to buffering by both protein and RNA chaperones. For example, the former might rescue misfolded proteins, whereas the latter removes translational roadblocks that predispose to misfolding. Elucidating to what extent protein and RNA chaperones have orthogonal buffering capacities will therefore be an important future objective.

Third, as is the case for protein chaperones, buffering by RNA chaperones is almost certain to occur through a range of mechanisms, so that identifying general principles of buffering in the face of mechanistic plurality will be a key challenge. Our findings should provide a strong impetus to meet this challenge and embark on further investigations with the ultimate aim to unravel the ramifications of mechanistically diverse chaperoning activity for RNA biogenesis, evolution, and evolvability.

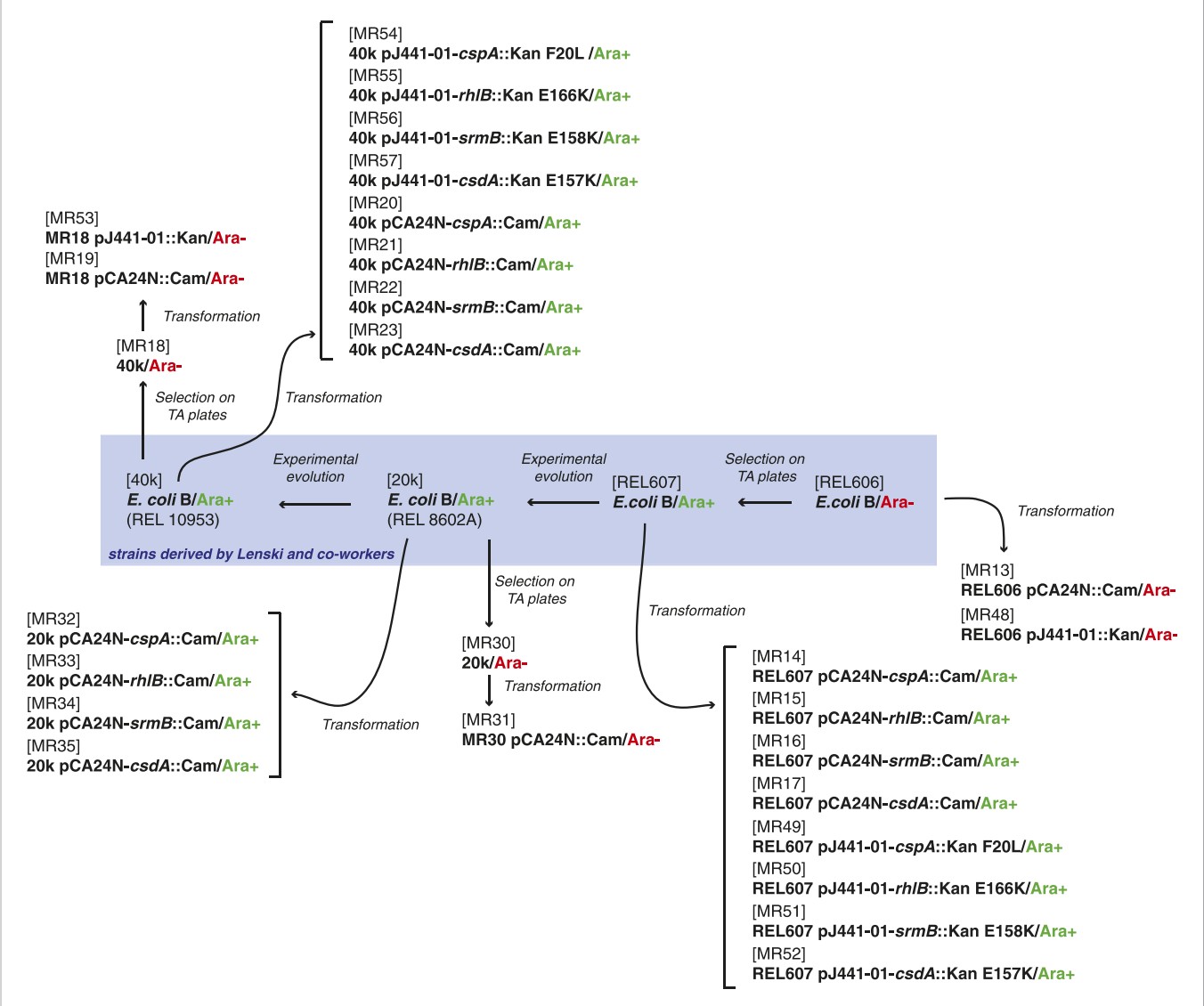

**Figure 6**. Relationship between REL606-derived strains.

Encouragingly, recent advances in probing RNA secondary structures and RNA–protein interactions at high throughput have rendered the transcriptome-wide dissection of RNA chaperone-mediated buffering a realistic prospect for the not too distant future.

## Materials and methods

### Bacterial strains, plasmids, and growth conditions

The strains used here were derived either from the *E. coli* K12 MG1655 strain by laboratory evolution and P1 transduction and/or transformation, or from the REL606 strain and its descendants in the LTEE (*Sniegowski et al., 1997*). All strains used are listed in *Supplementary file 4*, and their relationships and experimental derivation are illustrated in *Figures 6, 7*. Sequences of *cspA*, *rhlB*, *srmB*, and *csdA* inserted into pCA24N::Cam were obtained from the ASKA collection (http://www.shigen.nig.ac.jp/ecoli/strain/). For strain construction and subsequent experiments, bacteria were grown in LB at 37°C. To distinguish competitors during competition assays, cells were plated onto TA solid medium (*Lenski, 1991*).

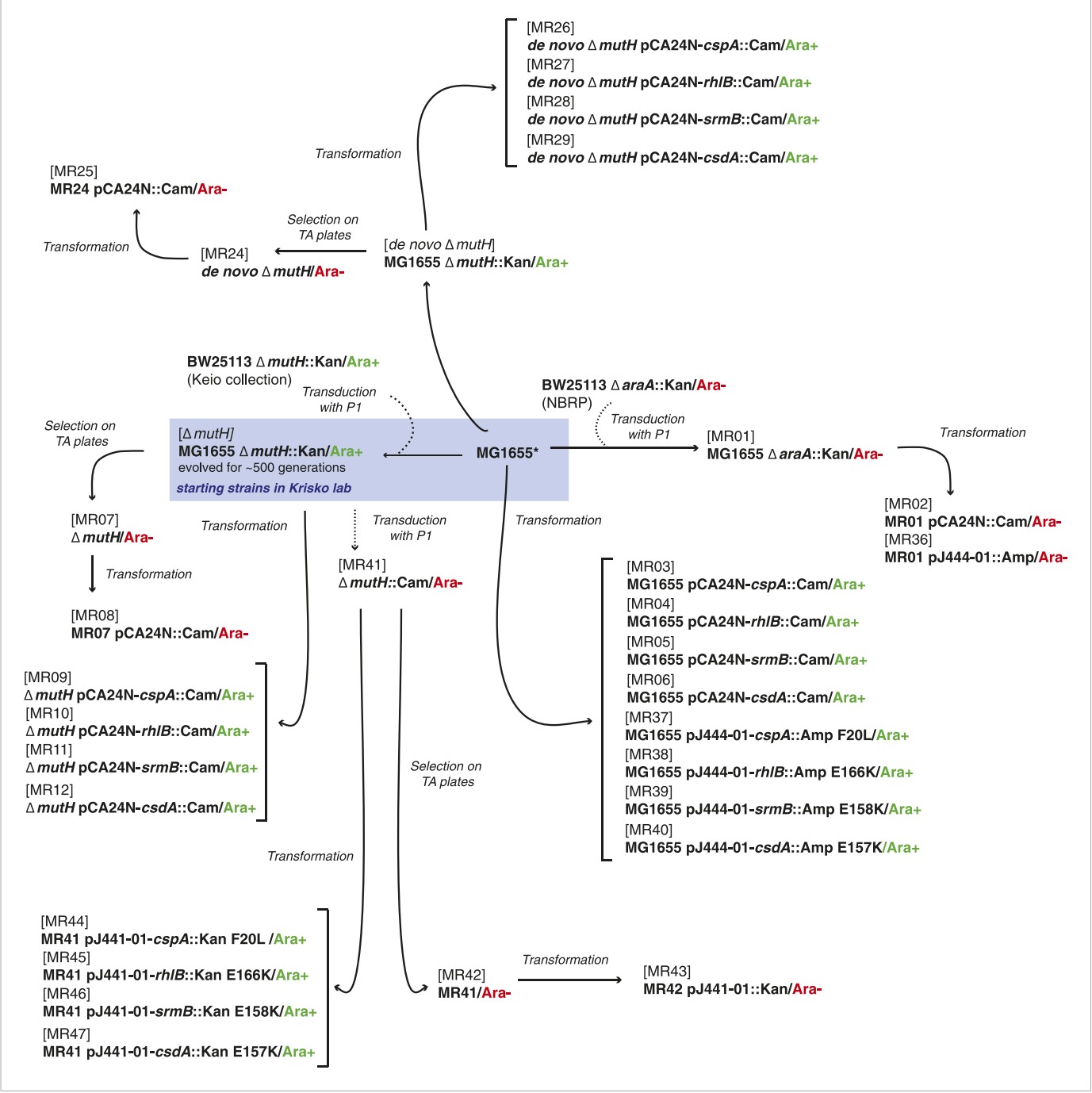

**Figure 7**. Relationship between MG1655-derived strains. *Resequencing of the MG1655 laboratory strain revealed a single difference to the NC000913 reference genome, an intergenic dinucleotide insertion at position 4296380 (AC → ACGC).

## Mutant proteins

Mutations in the DEAD domain of *E. coli* DBRHs abolish or severely reduce helicase activity, as demonstrated for the *E. coli* DBRHs RhlB (*Vanzo et al., 1998*), DbpA (*Elles and Uhlenbeck, 2008*), and CsdA (*Turner et al., 2007*). Here, we used DBRH mutants in which the central glutamic acid residue has been recoded to yield lysine, a change known to abolish RhlB ATPase activity, which is required for helicase activity (*Vanzo et al., 1998*). Mutations in the nucleic acid-binding domain of CspA were previously evaluated for their impact on both nucleic acid binding and protein

stability (*Hilier et al., 1998*). The F20L mutation was found to only weakly affect protein stability but strongly reduce nucleic acid binding (*Hilier et al., 1998*) and was therefore chosen for our study. Plasmids carrying the mutated genes were constructed by and purchased from DNA2.0.

## Competition assays

We performed pairwise competition experiments to estimate the relative fitness of two competing *E. coli* strains as previously described (*Lenski, 1991*). Briefly, the two competitors were grown separately, mixed at an initial ratio of 1:1 and diluted 100-fold in the competition environment (LB supplemented with the relevant antibiotics for plasmid maintenance). Initial and final densities (after either 24 or 2 hr) were estimated by diluting and spreading the cells on indicator TA (tetrazolium and arabinose) plates, which allow the competitors to be distinguished through an arabinose-utilization marker, which is neutral under the conditions utilized (*Lenski, 1988*). Relative fitness was calculated as

$$w = ln(A_f/A_i)/ln(B_f/B_i),$$

where *A* and *B* are the densities of the two competitors, and *i* and *f* represent initial and final densities, respectively. We used one-sample *t*-tests to evaluate whether mean relative fitness differed from the null expectation of one.

## Genome sequencing

DNA was extracted from the MG1655 ancestor and the evolved Δ*mutH* strain using the standard phenol–chloroform extraction procedure followed by ethanol precipitation. Libraries (100bp paired end reads) were prepared using the TruSeq DNA PCR-Free LT Sample Prep Kit (Illumina, San Diego, CA), with median insert size of 487 bp (ancestor) and 495 bp (evolved Δ*mutH*), for subsequent sequencing on the Illumina HiSeq 2000 platform using the TruSeq v3 reagent kit. Sequencing yielded 7365202 and 6504588 read pairs for the ancestral and evolved strains, respectively. The genomes of the REL606-derived 20k (REL8602A) and 40k (REL10953) clones were previously sequenced on the Illumina Genome Analyzer platform using a single lane of single-end 36bp reads per genome.

## Mutation identification

To identify genomic differences in the MG1655-derived strains, we closely followed the approach adopted by *Tenaillon et al. (2012)*. To detect single-nucleotide polymorphisms (SNPs) and short indels, paired end reads were first aligned to the reference genome (NC000913) using bwa mem (version 0.7.6a, arXiv: 1303.3997). Subsequently, duplicate reads were removed using samtools (version 0.1.19), and only those with non-zero mapping quality and no suboptimal alignments were considered further. SNPs and indels were then called using the mpileup function in samtools, requiring a minimum base quality of 30. Mutations were validated by visual inspection of read mappings in the Integrated Genome Viewer. Non-reference alleles had to be present at a frequency of $p > 0.75$ to be considered bona fide mutations. To ensure that we did not miss pertinent SNPs/indels in non-unique regions present in structural RNAs, we repeated the above procedure without filters on mapping quality and including suboptimal alignments. We identified no additional candidate mutations. Screening for larger deletions and duplications was performed by computing per-base coverage using the genomeCoverageBed function in bedtools (version 2.17.0) (*Quinlan and Hall, 2010*), subsequent smoothing across 250bp windows, normalizing by GC content, and applying a rolling filter to control for long-range effects, as previously described (*Tenaillon et al., 2012*). This approach confirmed the *mutH* deletion in the mutator strain. We found no other large deletions. Similarly, no duplications were detected after visually inspecting neighbouring regions with unusual differences in coverage (>1.5-fold). All candidate regions were found to be due to local drop-offs in neighbouring regions rather than excess coverage in focal windows. For REL606-derived strains, candidate genomic differences compared with the ancestral genome were identified using the PALOMA (*Vallenet et al., 2012*) and BRESEQ (*Barrick et al., 2009*) pipelines. Sequencing data for the MG1655-derived strains and the 20k strain have been deposited in the European Nucleotide Archive (accession no. PRJEB7107). Sequencing data for the REL606-derived 40k clone was previously deposited in the National Center for Biotechnology Information Sequence Read Archive (accession no. SRR1536189).

## Introduction of point mutations into ancestral genetic backgrounds

All point mutations detected in the evolved Δ*mutH* strain that were located in a gene of known function (i.e., not located in a y-gene) were individually introduced into the ancestral MG1655

background using a recombineering approach (*Sawitzke et al., 2013*). Briefly, we designed single-stranded oligonucleotides, ~70 nt in length, that were complementary to the region of interest and carried the desired point mutation (*Supplementary file 5*). They were transformed by electroporation into the TB56 (Ara+) and TB62 (Ara−) strains supplemented with pSIM6. In both TB56 and TB62 (kindly provided by Tobias Bergmiller), the native *mutS* promoter has been replaced with an *ara* promoter. In the presence of 0.2% arabinose, the strains have wild-type MutS levels, whereas in the presence of 0.2% glucose, MutS expression is repressed. Growing strains in LB medium supplemented with 0.2% glucose therefore increases the likelihood that oligo-born mutations are fixed due to impaired mismatch repair. Following electroporation, cells were grown overnight at 32°C on LB agar plates supplemented with 0.2% arabinose, and the presence of mutations was confirmed by sequencing target regions from individual colonies on an ABI PRISM 310 Genetic Analyzer using the Big Dye Terminator 1.1 Cycle Sequencing kit (Life Technologies, Carlsbad, CA). The primers used for sequencing are listed in *Supplementary file 5*.

## Determination of relative chaperone levels

Exponentially growing evolved Δ*mutH* strains overexpressing one of the four RNA chaperone proteins (CspA, RhlB, SrmB or CsdA) were pelleted by centrifugation and resuspended in UTC$^{DTT}$ buffer containing 8 M urea, 2 M thiourea, 4% CHAPS and 10 mM DTT supplemented with a mixture of protease inhibitors containing aprotinin bestatin, leupeptin, pepstatin A, E-64 and AEBSFxHCL, EDTA-free (Life Technologies). Cells were incubated for 2 hr at room temperature followed by a 20-min centrifugation step at 12,000×*g*. Protein concentrations were determined using the Bradford assay (*Bradford, 1976*). For each sample, 15 µg of total protein extract was loaded onto an SDS-PAGE gel with a 6% stacking and 20% resolving gel. 6X-His tagged CspA, RhlB, SrmB, and CsdA were detected by Western blotting using a mouse monoclonal anti-6X His tag antibody (Abcam, United Kingdom) followed by a goat anti-mouse polyclonal antibody conjugated to HRP (Abcam). Proteins were visualized on autoradiographic film using the Amersham ECL Advance chemiluminescence detection system (GE Healthcare Life Sciences, United Kingdom). We used ImageJ (*Collins, 2007*) to quantify the intensity of each chaperone band on the Western blot and normalized this intensity by the amount of total protein loaded into each lane (detected by Coomassie staining of the SDS-PAGE gel and subsequent quantification with ImageJ). This normalized abundance allows comparing relative chaperone levels across experiments (*Figure 5C*).

## Acknowledgements

We thank Heinz Himmelbauer and the CRG Genomics Unit for sequencing the Δ*mutH* mutator strain and its laboratory ancestor; Sanjay Khadayate for input into mutation identification; and Laurence Hurst, Ben Lehner, Fran Supek, and Bojan Zagrovic for comments. MR and AK are grateful to Tea Copic for technical support and to Tobias Bergmiller for the TB56 and TB62 strains and valuable advice. This work was funded by the European Union program FP7-ICT-2013-10 project EvoEvo grant 610427, the Université Grenoble Alpes and the CNRS (to DS); MRC core funding (to TW); Fondation Nelia et Amedeo Barletta and the Mediterranean Institute for Life Sciences (to MR and AK).

## Additional information

### Funding

| Funder | Grant reference | Author |
| --- | --- | --- |
| European Commission | FP7-ICT-2013-10 project EvoEvo grant 610427 | Dominique Schneider |
| Universite Grenoble Alps | | Dominique Schneider |
| Centre National de la Recherche Scientifique | | Dominique Schneider |
| Medical Research Council (MRC) | | Tobias Warnecke |

| Funder | Grant reference | Author |
|---|---|---|
| Fondation Nelia et Amadeo Barletta | | Marina Rudan, Anita Krisko |
| Mediterranean Institute for Life Sciences | | Marina Rudan, Anita Krisko |

The funders had no role in study design, data collection, and interpretation, or the decision to submit the work for publication.

## Author contributions

MR, performed the experiments and contributed to experimental design and writing of the article; DS, provided the REL606-derived strains and sequences and contributed to the writing of the article; TW, conceived the study, designed experiments, analyzed the data, provided the MG1655-derived sequences and contributed to the writing of the article; AK, designed experiments, analyzed the data, and contributed to the writing of the article

## Additional files

### Supplementary files

• Supplementary file 1. Mutations in the 20k strain.

• Supplementary file 2. Mutations in the 40k strain.

• Supplementary file 3. Mutation in the evolved Δ*mutH* strain.

• Supplementary file 4. List of *Escherichia coli* strains used in this study.

• Supplementary file 5. Primers used for recombineering and sequencing.

### Major datasets

The following datasets were generated:

| Author(s) | Year | Dataset title | Dataset ID and/or URL | Database, license, and accessibility information |
|---|---|---|---|---|
| Rudan M, Schneider D, Warnecke T, Krisko A | 2014 | RNA chaperones buffer deleterious mutations in E. coli | http://www.ebi.ac.uk/ena/data/view/PRJEB7107 | Publicly available at the EBI European Nucleotide Archive under study accession PRJEB7107. |
| Raeside C, Gaffé J, Deeatherage DE, Tenaillon O, Briska AM, Ptashkin RN, Cruveiller S, Médigue C, Lenski RE, Barrick JE, Schneider D | 2014 | Large Chromosomal Rearrangements during a Long-Term Evolution Experiment with Escherichia coli | http://www.ncbi.nlm.nih.gov/sra/?term=SRR1536189 | Publicly available at the NCBI Sequence Read Archive (SRR1536189). |

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
