## [Decision Letter]

Thank you for sending your work entitled “RNA chaperones buffer deleterious mutations in *E. coli*” for consideration at *eLife*. Your article has been favorably evaluated by Detlef Weigel (Senior editor), a Reviewing editor, and two reviewers.

The Reviewing editor and the reviewers discussed their comments before we reached this decision, and the Reviewing editor has assembled the following comments.

The conclusion of the paper is highly interesting: that RNA chaperones may buffer mutations that could impact on mRNA folding. The possible implication is that this way hidden genetic variability can be maintained in the genome and become effecting under certain (e.g. stress) conditions by compensating the function of RNA chaperones, e.g. by over-loading their functions. This role of chaperone was previously reported and extensively discussed in the context of protein mutations, but not in the context of RNA mutation.

Stronger evidence that this is indeed the case, however, is required. The main issue raised is that the mutations described were not characterized in the context of RNA stability. Therefore, it is difficult to interpret the experiment. If you decide to submit a substantially revised version of the manuscript, a minimal set of experiments required to better support your conjecture has to include: (1) the identification of at least one mutation in the original strain predicted to affect RNA stability (using computational tools), (2) the demonstration that it has a detrimental effect on fitness when in isolation, and (3) evidence that it is being buffered by RNA chaperones.

Additional comments:

1) Given the overall conclusion of the work, that overexpression of RNA chaperone proteins confers a fitness benefit by assisting with RNA folding or structural rearrangements in the mutated strains, it seems important to evaluate whether any of the mutations in known or suspected structured RNAs. This question is especially intriguing for the *ΔmutH* because it has such a small number of mutations, and all of the SNPs are in coding sequence. This could be done computationally.

2) The data presentation does not allow a direct comparison, but the impression upon comparing parts B and C of Figure 1 is that, upon over-expression of some DBRHs , the evolved 40k strain grows faster than the ancestral strain (RhlB and CsdA; or the latter and SrmB, in Figure 2). Is that the case? If so, how can this be accounted for?

3) Fitness was determined by competitions: cultures were grown for 24 hours, as default. It's unclear to which growth phase the fitness effects relate to: is it most exponential growth, or stationary phase as well. Further, given the different growth rates of the different strains, it could may well be that some were subjected to growth only and other to growth + stationary phase survival. Many, I'd say, most mutations have conflicting effects on growth vs. survival in stationary phase. It could therefore be that the actual buffering effects would be different, possibly much larger, if competitions were perfumed under a defined phase—exponential growth on one hand, and survival at stationary phase on the other. This should be clarified by performing competition experiment in well-defined conditions (e.g. maintaining cells in log phase).

4) “Relative fitness was calculated as the ratio of the competitors' growth rates during competition.” The method relates to counting the ratio of CFUs on tetrazolium plates, so how would growth rates be derived from that; especially, when no data is available as number of generations within the 24 hours growth (or growth phase, see above)?

5) “The arabinose-utilization marker, which is neutral under the conditions utilized”. Was this tested, or assumed? Many markers do affect growth rates.

6) An empty vector control is missing.

7) The supplementary tables listing the mutations are not very informative. The genes and positions of the mutations should be specified.

8) The expression levels of the RNA chaperones are not reported. It is interesting that overexpression of CspA restores fitness to the same level as overexpression of the ATP-dependent DEAD-box proteins, but this result cannot be fully evaluated without knowing whether the different proteins were expressed at comparable levels.

9) The sentence: “RNA chaperones can promote orderly structural transitions towards and subsequently stabilize the native fold or—as exemplified by classic work on the *Neurospora crassa* CYT-19 protein—facilitate the re-folding of misfolded species (reviewed in Bogumil, 2010).” This reference seems to be incorrect here.

10) In the subsection headed “Mutant proteins”, the sentence: “Here, we used DBRH mutants in which the central glutamic acid residue has been recoded to yield lysine, a change known to abolish RhlB helicase activity (Vanzo, 1998).” In Vanzo, 1998, helicase activity was not measured. Instead, it was shown that RNaseE-dependent ATPase activity is compromised for this mutant. It is certainly expected that the mutation will also result in a loss of helicase activity, which depends on ATPase activity, but the connections to previous work should be clarified.

[Editors' note: further revisions were requested prior to acceptance, as described below.]

Thank you for resubmitting your work entitled “RNA chaperones buffer deleterious mutations in *E. coli*” for further consideration at *eLife*. Your revised article has been favorably evaluated by Detlef Weigel (Senior editor), a Reviewing editor, and two reviewers. The manuscript has been improved but there are some remaining issues that need to be addressed before acceptance, as outlined below:

As you can see from the reviews, both reviewers appreciated the additional data and discussion provided in the revised manuscript and agreed that the paper is improved. There is also an agreement that you nicely illustrated the core finding—that the slow growth phenotypes of these strains can be rescued with overexpression of DEAD-box proteins, known RNA-dependent ATPases (and one RNA-binding protein that lacks ATPase activity)—and that this finding is interesting.

However, both reviewers also agreed that the mechanism(s) of rescue is not at all understood, and that you haven't convincingly proved that it has to do with buffering mRNA stability.

What we would like to ask you is to relate to these points specifically in the manuscript. Currently, the Discussion is very short, and it can easily be extended to a real section where you can discuss and relate to the all the points raised by the reviewers (in particular the points raised by reviewer #1).

It may also be interesting (but not necessary) to relate to protein chaperones. Would you expect mutations that would also be buffered this way at the protein level? What do you think would be more important?

*Reviewer #1*:

The authors have done a significant amount of work, and provided new data. The overall picture remains, however, enigmatic, and possibly even more confusing than it had been prior to the addition of these new data. Specifically, there is still no clear indication that the mutations in the evolved lines relate to RNA stability/folding, and are thereby buffered by RNA chaperones.

The puzzling points that cast doubt on this interpretation are several:

1) There is not even a single overlapping mutation between the two evolved lines. This may suggest a non-specific buffering mechanism, but then on the other hand, as with protein chaperones, RNA chaperones are likely to have some obligatory 'clients', and other RNAs that are highly dependent on chaperone action. However, none of the identified mutations is in known structural RNAs.

2) Making isogenic strains is probably the most rigorous test possible, and the authors took this avenue. Sadly, however, the results are not supporting the hypothesis. Only one out 7 mutations tested (in *lamB*) exhibited both reduced fitness and evidence for buffering. But this mutation involves an amino acid exchange.

3) The synonymous mutation in *rplS* is more convincing, as it was shown to be deleterious and compensated by DRBH over-expresssion. However, then authors' assumption that fitness effects of synonymous mutations indicate the presence of selective constraints beyond the protein level is wrong. Synonymous mutations have been shown to affect the rate and outcome of protein folding, and thus the yield of soluble, function protein.

Overall, as I said, I'm sorry to say that the additional data cast in my view more doubts on the paper's main conclusion.

*Reviewer #2*:

The authors have addressed my major concerns. The work demonstrates that mutations in the evolved strains are reduced in growth and this reduction can be 'buffered' by overexpression of proteins that function as ATP-dependent or ATP-independent RNA chaperones. It is interesting that both mutations that confer fitness defects individually are in coding regions. While one is a missense mutation, the other one is synonymous, suggesting a role of the chaperone proteins in regulating the lifetime or translatability of the mRNA. The work raises very interesting questions on the roles of RNA chaperones in the functions of mRNAs, which will presumably spur future work.

---

## [Author Response]

*The main issue raised is that the mutations described were not characterized in the context of RNA stability. Therefore, it is difficult to interpret the experiment. If you decide to submit a substantially revised version of the manuscript, a minimal set of experiments required to better support your conjecture has to include: (1) the identification of at least one mutation in the original strain predicted to affect RNA stability (using computational tools), (2) the demonstration that it has a detrimental effect on fitness when in isolation, and (3) evidence that it is being buffered by RNA chaperones*.

We now describe two mutations, one in each ancestral genotypic background, that have a deleterious effect in isolation and whose effect on fitness is buffered by RNA chaperone overexpression. The mutations are located in the ribosomal protein gene *rplS* (REL606) and the glycoporin-encoding gene *lamB* (MG1655), respectively. Interestingly, the mutation in *rplS* is synonymous, ruling out protein-level effects, and fitness defects are consistent with previous observations for *rplA* and *rpsT* in *Salmonella typhimurium* ([21] Science 330:825). We discuss likely pathways to fitness loss for both mutations (in the Results and Discussion section) along with predictions of how mutations might affect RNA secondary structure.

Rather than attempt and predict likely candidates for buffering computationally (a hard problem, see below), we took advantage of the small number of point mutations in the evolved Δ*mutH* strain to pinpoint fitness-relevant mutations systematically. Using a recombineering approach, we introduced individual mutations into the ancestral MG1655 background and competed these strains, isogenic except for a single mutation, with MG1655. Of the 12 point mutations in the evolved Δ*mutH* strain, we tested 7, comprising all mutations located in genes of known function.

For the 40k strain, where a systematic approach is prohibited by the large number of mutations, we considered phenotypic, expression, and structural information along with prior literature, to make what is best described as an educated guess regarding which mutation might compromise fitness. As alluded to above, making systematic computational predictions here is non-trivial. First, neither strain bore mutations in known structural RNAs (also see point 1 below), so we lacked obvious candidates for buffering. In fact, all mutations in the evolved Δ*mutH* strain were located in coding regions of nonessential genes, where it is difficult to assess the biological significance of RNA structural change. Second, existing computational tools can model mutational effects on RNA secondary structure, but provide no insight into whether a given mutation can be rescued by RNA chaperones. Third, it is not obvious how to extrapolate from molecular fitness, i.e. how disruptive a mutation is at the RNA structural level, to organismal fitness. For example, a relatively small change in structure in a highly expressed gene might matter more than a strongly disruptive mutation in a gene that is not even expressed. Given the likely complex determinants of whether rescue can occur, we would have needed a much larger set of deleterious mutations amenable to buffering to train and validate a predictive algorithm. Thus, while we provide multiple metrics of RNA structural change ([Supplementary-material SD1-data SD2-data SD3-data]), some of which do indeed suggest that the deleterious mutations we identify are associated with structural change of above average severity, we believe that these correlations should be treated with a healthy amount of caution. This is why we opted to pursue the systematic experimental approach described above.

*Additional comments*:

1) *Given the overall conclusion of the work, that overexpression of RNA chaperone proteins confers a fitness benefit by assisting with RNA folding or structural rearrangements in the mutated strains, it seems important to evaluate whether any of the mutations in known or suspected structured RNAs. This question is especially intriguing for the ΔmutH because it has such a small number of mutations, and all of the SNPs are in coding sequence. This could be done computationally*.

As previously mentioned in the Methods section, neither strain harbours mutations in known structural RNAs (tRNAs, rRNAs). We now highlight this more prominently in the main text (in the subsection headed “Pinpointing individual deleterious mutations buffered by RNA chaperones ”). Indeed, we took extra care with SNP calling in those regions, because a lack of unique read mapping might have concealed the presence of pertinent mutations. This was not the case (see Methods).

We further considered whether mutations might be located in experimentally described structured elements of known functional importance, such rho-independent terminators or REP elements. However, surprisingly, all point mutations in the evolved *ΔmutH* strain were located in the coding regions of nonessential genes. This is problematic when it comes to computational predictions of phenotypic effect: all mRNAs will be predicted to exhibit a multitude of local structures but for the vast majority of individual structures, we have no idea whether they are functionally relevant. As elaborated earlier, we considered computational predictions here to be of limited utility and instead decided to systematically probe mutations experimentally for this genetic background.

2) *The data presentation does not allow a direct comparison, but the impression upon comparing parts B and C of*
Figure 1
*is that, upon over-expression of some DBRHs , the evolved 40k strain grows faster than the ancestral strain (RhlB and CsdA; or the latter and SrmB, in*
Figure 2*). Is that the case? If so, how can this be accounted for*?

It is not so much data presentation than the nature of the data that precludes a valid comparison. We think it is important to highlight that competition assays are designed to quantify the relative fitness of different strains competing in the same environment. They do not allow extrapolation of how strain A in one competition would perform against strain B in another experiment. This is exacerbated in the current case: in Figure 1 two different genotypes are competed with each other; in Figure 1 the genotype is the same but one strain carries an overexpression plasmid, whereas the other carries an empty plasmid.

From a theoretical perspective, we think it is possible in principle (although perhaps unlikely) that a 40k overexpression strain outcompetes a REL606 overexpression strain (the experiment that would address the referee’s suspicion directly). For example, some evolved changes in the 40k genotype might incidentally have eliminated potential costs associated with overexpression in REL606. This theoretical possibility is another reason why experiments in Figure 1 should not be compared directly.

3) *Fitness was determined by competitions: cultures were grown for 24 hours, as default. It's unclear to which growth phase the fitness effects relate to: is it most exponential growth, or stationary phase as well. Further, given the different growth rates of the different strains, it could may well be that some were subjected to growth only and other to growth + stationary phase survival. Many, I'd say, most mutations have conflicting effects on growth vs. survival in stationary phase. It could therefore be that the actual buffering effects would be different, possibly much larger, if competitions were perfumed under a defined phase—exponential growth on one hand, and survival at stationary phase on the other. This should be clarified by performing competition experiment in well-defined conditions (e.g. maintaining cells in log phase)*.

The 24h competitions were designed to assay fitness across a full growth cycle in part because we expected to find and wanted to capture small differences in fitness, which might only become evident after extended periods of competition. As a consequence, the fitness we report is a compound measure that subsumes the contributions of individual mutations and how these contributions vary across the course of the experiment (i.e. with growth phase). We agree that it would, in addition, be interesting to describe fitness effects in a more fine-grained fashion, establishing which mutations matter when. We therefore followed the referees’ suggestion and performed an additional set of competition experiments in which, instead of running the competition for 24 hours, we diluted and spread the cells on indicator TA plates after 2 hours, when cultures are in mid-exponential phase.

We make the following observations (presented in figure supplements to Figures 1, 2, 3 and 4): First, the 40k and evolved *ΔmutH* strains exhibit lower fitness compared to their respective ancestor, whereas the 20k and de novo *ΔmutH* strains do not. This demonstrates that, in both evolved strains, there are mutations that have fitness-relevant effects during the early growth phases (up to mid-exponential). Second, 40k/evolved *ΔmutH* strains that overexpress one of the RNA chaperones outcompete strains of the same genotype that carry an empty control plasmid, suggesting that mutations with early-growth-phase relevance are being buffered by chaperone overexpression. For most chaperone/genotype combinations there are quantitative differences in relative fitness after 2h and 24h, consistent with the possibility that differential survival during later growth can further exacerbate or ameliorate fitness differences. It is interesting to note that buffering in the evolved *ΔmutH* strain is consistently stronger after 24h whereas the picture is more variable in the 40k strain. We suggest (in the Results and Discussion section) this might reflect a larger number of contributory mutations in the 40k strain. This suggestion is further supported by the observation that competitions involving the isolated *lamB* mutation broadly replicate overall trends for the evolved *ΔmutH* strain (i.e. smaller fitness differences in exponential phase), in line with this mutation being a strong contributor to overall fitness effects. In contrast, fitness effects of the *rplS* mutation introduced in isolation into the ancestor of the 40k strain does not track overall fitness patterns for this strain, again consistent with the presence of multiple fitness-relevant mutations in this strain.

Finally, on a technical note, we believe that reduced variability between biological replicates in the 2h compared to the 24h experiments is likely owing to the fact that pipetting a similar number of cells is easier under the more diluted conditions after 2h, although we cannot at this point rule out that greater stochasticity is biological in origin.

*4)* “*Relative fitness was calculated as the ratio of the competitors' growth rates during competition.*” *The method relates to counting the ratio of CFUs on tetrazolium plates, so how would growth rates be derived from that; especially, when no data is available as number of generations within the 24 hours growth (or growth phase, see above)*?

Relative fitness was measured as:w=ln(Af/Ai)/ln(Bf/Bi),

where A and B are the densities of the two competitors and *i* and *f* represent initial and final densities, respectively.

This calculation reflects the differences between the two competitors in lag, growth, and stationary phases over the same serial-transfer cycle. This formula has been applied previously to measure relative fitness, notably in the long-term evolution experiment. We have revised the text and included the above formula (in the subsection headed “Competition assays”) to make this computation more explicit.

*5)* “*The arabinose-utilization marker, which is neutral under the conditions utilized*”*. Was this tested, or assumed? Many markers do affect growth rates*.

Neutrality of the arabinose-utilization marker was previously demonstrated by Lenski in the context of establishing the long-term evolution experiment (Lenski, R.E., 1988). We have added this reference to support our original statement (in the subsection headed “Competition assays”).

*6) An empty vector control is missing*.

An empty vector control is already present wherever appropriate. Competition experiments have to be controlled internally and all strains bearing a chaperone overexpression vector are therefore competed against the same genotype bearing an empty vector. We highlight this in the text (in the subsection headed “Overexpression of DEAD box RNA helicases enhances fitness of a low-fitness strain”) and in all relevant figures and their legends.

7) *The supplementary tables listing the mutations are not very informative. The genes and positions of the mutations should be specified*.

We agree that the original tables were rather Spartan in information content. We have revised all supplementary files to provide salient annotations to help the reader place mutations in biological context. The revised tables ([Supplementary-material SD1-data SD2-data SD3-data]) detail not only in/near which gene a particular mutation is located, but also its effect on the protein sequence (if any), the expression level of that gene in LB and minimal medium, essentiality status, COG functional class, and a number of properties relating to the difference between predicted WT and mutant RNA secondary structures. Not all these features are explicitly discussed in the text, but we decided to include them in the table for convenient reference.

8) *The expression levels of the RNA chaperones are not reported. It is interesting that overexpression of CspA restores fitness to the same level as overexpression of the ATP-dependent DEAD-box proteins, but this result cannot be fully evaluated without knowing whether the different proteins were expressed at comparable levels*.

We now provide a semi-quantitative assessment of relative chaperone levels in the evolved Δ*mutH* overexpression strains. Briefly, we expressed His-tagged versions of the four RNA chaperones (*csdA, rhlB, srmB, cspA*), quantified expression levels from Western blots and normalized by the intensity of Coomassie-stained total protein (see Methods and Results). This allows an assessment of relative chaperone levels in the different overexpression strains. For example, as indicated in the new Figure 5, RhlB levels in RhlB-overexpressing cells are ∼2-fold higher than CspA levels in CspA-overexpressing cells. Note that this metric does not allow conclusions about the absolute fraction of total protein that is occupied by each chaperone in the different strains. Overall, relative expression levels vary ∼4-fold, with CspA exhibiting the lowest relative expression levels following overexpression. As noted in the text (in the subsection headed “Fitness gains upon overexpression of *cspA* suggest diverse mechanisms of buffering”), this might be related to known negative autoregulation of *cspA* ([1], JBac 179:7081).

In the future, it will be interesting to build on these preliminary results and provide a more comprehensive characterization of protein levels across the numerous strains in this study, with the ultimate aim to elucidate how buffering efficacy differs across RNA chaperones.

9) *The sentence:* “*RNA chaperones can promote orderly structural transitions towards and subsequently stabilize the native fold or—as exemplified by classic work on the Neurospora crassa CYT-19 protein—facilitate the re-folding of misfolded species (reviewed in Bogumil, 2010).*” *This reference seems to be incorrect here*.

Thank you for catching this error. We have replaced the incorrect reference with the one we meant to use ([30], Front Biosci 13, 1–20).

10) *In the subsection headed “Mutant proteins”, the sentence:* “*Here, we used DBRH mutants in which the central glutamic acid residue has been recoded to yield lysine, a change known to abolish RhlB helicase activity (Vanzo, 1998).*” *In Vanzo, 1998, helicase activity was not measured. Instead, it was shown that RNaseE-dependent ATPase activity is compromised for this mutant. It is certainly expected that the mutation will also result in a loss of helicase activity, which depends on ATPase activity, but the connections to previous work should be clarified*.

We have clarified the link between DEAD motif mutants and ATPase/helicase activity (in the subsection headed “Mutant proteins”), We have clarified the link between DEAD motif mutants and ATPase/helicase activity (in the subsection headed “Mutant proteins”). As mentioned by the referee, mutating the DEAD motif directly affects ATPase activity. However, since helicase activity is dependent on ATPase activity, DEAD motif mutants also compromise helicase activity.

[Editors' note: further revisions were requested prior to acceptance, as described below.]

*As you can see from the reviews, both reviewers appreciated the additional data and discussion provided in the revised manuscript and agreed that the paper is improved. There is also an agreement that you nicely illustrated the core finding—that the slow growth phenotypes of these strains can be rescued with overexpression of DEAD-box proteins, known RNA-dependent ATPases (and one RNA-binding protein that lacks ATPase activity)—and that this finding is interesting*.

*However, both reviewers also agreed that the mechanism(s) of rescue is not at all understood, and that you haven't convincingly proved that it has to do with buffering mRNA stability*.

*What we would like to ask you is to relate to these points specifically in the manuscript. Currently, the Discussion is very short, and it can easily be extended to a real section where you can discuss and relate to the all the points raised by the reviewers (in particular the points raised by reviewer #1)*.

We are gratified that the reviewers agree that our core finding—that deleterious mutations can be buffered by RNA chaperones—is strongly supported by the evidence. This is the key claim of the paper. We agree with the reviewers that the specific mechanism(s) of buffering remain enigmatic. In fact, we were very careful not to make mechanistic claims, although we naturally discussed the modus operandi of DEAD box RNA helicases and CspA.

Although this study was focused on demonstrating buffering at the organismal level rather than on elucidating the molecular mechanism(s) of buffering, we appreciate the desire for a more extensive discussion of the latter. Since our experiments cannot directly inform on mechanism, we have focused on illustrating (rather than exhaustively enumerating) potential buffering pathways that are consistent with the data while highlighting more general considerations that should guide follow-up investigations. This includes considerations of the significance of synonymous and non-synonymous mutations and whether fitness effects are proximately mediated through protein defects. We also now highlight that the individual deleterious mutations we identify will provide an invaluable platform for the mechanistic dissection of buffering in the future.

Reviewer #1:

*The authors have done a significant amount of work, and provided new data. The overall picture remains, however, enigmatic, and possibly even more confusing than it had been prior to the addition of these new data. Specifically, there is still no clear indication that the mutations in the evolved lines relate to RNA stability/folding, and are thereby buffered by RNA chaperones*.

It is important to discriminate two logically distinct claims here, one being that RNA chaperones buffer deleterious mutations (a claim about fitness that we make and provide strong evidence for), the other being that buffering occurs via a specific mechanism that involves certain changes in RNA stability and/or folding (a claim about mechanism that we are keenly aware we cannot make based on the data presented). The new experiments have indeed not established a specific mechanism (doing so will require further work), but their discovery has enabled mechanistic explorations to be carried out in the future, which is a significant advance in itself.

*The puzzling points that cast doubt on this interpretation are several*:

1) *There is not even a single overlapping mutation between the two evolved lines. This may suggest a non-specific buffering mechanism, but then on the other hand, as with protein chaperones, RNA chaperones are likely to have some obligatory 'clients', and other RNAs that are highly dependent on chaperone action. However, none of the identified mutations is in known structural RNAs*.

Mutations in structural RNAs are expected to be exceedingly rare in evolving colonies as fitness effects tend to be severe enough for mutants to quickly get outcompeted. But finding mutations outside of known structural RNAs does not invalidate a model where buffering occurs through RNA structural remodeling. More fundamentally, buffering (regardless of mechanism) clearly does occur and is dependent on helicase activity/nucleic acid binding. The unanswered question is not if RNA chaperones buffer mutations but how.

2) *Making isogenic strains is probably the most rigorous test possible, and the authors took this avenue. Sadly, however, the results are not supporting the hypothesis. Only one out 7 mutations tested (in lamB) exhibited both reduced fitness and evidence for buffering. But this mutation involves an amino acid exchange*.

The mutations we find cannot and do not conclusively implicate a specific buffering mechanism as we readily acknowledge in the manuscript. However, it is important to highlight that finding a mutation to be non-synonymous does not automatically disqualify it from consideration as causing an issue at the level of RNA structure. Both synonymous and non-synonymous sites affect mRNA folding. It simply places a heavier burden of proof on future experiments to show that either the effect on protein sequence is irrelevant or that it is relevant but buffering nonetheless mediated at the RNA level (as in a hypothetical scenario where the mutation has a dominant negative effect on the protein but RNA chaperone overexpression leads to beneficial degradation). We discuss these issues in the revised manuscript.

3) *The synonymous mutation in rplS is more convincing, as it was shown to be deleterious and compensated by DRBH over-expresssion. However, then authors' assumption that fitness effects of synonymous mutations indicate the presence of selective constraints beyond the protein level is wrong. Synonymous mutations have been shown to affect the rate and outcome of protein folding, and thus the yield of soluble, function protein*.

We now briefly discuss synonymous and non-synonymous mutations and their potential effects at the RNA and protein level. We have also removed the ambiguous term “selective constraints beyond the protein level” and replaced it with the more concise “selective constraints unrelated to amino acid composition”.